# A qualitative study of the barriers and enhancers to retention in care for pregnant and postpartum women living with HIV

John Humphrey[1]*, Marsha Alera[2], Bett Kipchumba[3], Elizabeth J. Pfeiffer[4], Julia Songok[5], Winfred Mwangi[3], Beverly Musick[6], Constantin Yiannoutsos[6], Juddy Wachira[7], Kara Wools-Kaloustian[1]

1 Department of Medicine, Indiana University School of Medicine, Indianapolis, IN, United States of America, 2 Academic Model Providing Access to Healthcare, Eldoret, Kenya, 3 Department of Reproductive Health, Moi Teaching and Referral Hospital, Eldoret, Kenya, 4 Department of Anthropology, Rhode Island College, Providence, RI, United States of America, 5 Department of Child Health and Paediatrics, College of Health Sciences, Moi University, Eldoret, Kenya, 6 Department of Biostatistics, Indiana University School of Medicine, Indianapolis, IN, United States of America, 7 Department of Behavioral Sciences, College of Health Sciences, Moi University, Eldoret, Kenya

* humphrjm@iu.edu

**Data Availability Statement:** The qualitative data illustrating the findings of the study are presented as participant quotes within the paper. The raw

## Abstract

Retention in care is a major challenge for pregnant and postpartum women living with HIV (PPHIV) in the prevention of mother-to-child HIV transmission (PMTCT) continuum. However, the factors influencing retention from the perspectives of women who have become lost to follow-up (LTFU) are not well described. We explored these factors within an enhanced sub-cohort of the East Africa International Epidemiology Databases to Evaluate AIDS Consortium. From 2018–2019, a purposeful sample of PPHIV ≥18 years of age were recruited from five maternal and child health clinics providing integrated PMTCT services in Kenya. Women retained in care were recruited at the facility; women who had become LTFU (last visit >90 days) were recruited through community tracking. Interview transcripts were analyzed thematically using a social-ecological framework. Forty-one PPHIV were interviewed. The median age was 27 years, 71% were pregnant, and 39% had become LTFU. In the individual domain, prior PMTCT experience and desires to safeguard infants' health enhanced retention but were offset by perceived lack of value in PMTCT services following infants' immunizations. In the peer/family domain, male-partner financial and motivational support enhanced retention. In the community/society domain, some women perceived social pressure to attend clinic while others perceived pressure to utilize traditional birth attendants. In the healthcare environment, long queues and negative provider attitudes were prominent barriers. HIV-related stigma and fear of disclosure crossed multiple domains, particularly for LTFU women, and were driven by perceptions of HIV as a fatal disease and fear of partner abandonment and abuse. Both retained and LTFU women perceived that integrated HIV services increased the risk of disclosure. Retention was influenced by multiple factors for PPHIV. Stigma and fear of disclosure were prominent barriers for LTFU women. Multicomponent interventions and refining the structure and efficiency of PMTCT services may enhance retention for PPHIV.

interview transcripts contain information that could potentially compromise participant privacy. Therefore, requests to access these data should be made in writing to the AMPATH Research Manager, Ms. Jepchirchir Kiplagat, at jkiplagat@ampath.or.ke.

**Funding:** This research was supported by the National Institute Of Allergy And Infectious Diseases (NIAID) and the Eunice Kennedy Shriver National Institute Of Child Health & Human Development (NICHD), in accordance with the regulatory requirements of the National Institutes of Health under Award Number U01AI069911 East Africa IeDEA Consortium (K.W., C.Y.). This research was also supported by the NICHD under Award Number K23HD105495 (J.H.). The content is solely the responsibility of the authors and does not necessarily represent the official views of the National Institutes of Health. The funders had no role in study design, data collection and analysis, decision to publish, or preparation of the manuscript.

**Competing interests:** The authors have declared that no competing interests exist.

## Introduction

Retention in care is a major challenge in the prevention of mother-to-child transmission of HIV (PMTCT) continuum in sub-Saharan Africa. Studies across the sub-continent have shown that nearly a third of women initiating antiretroviral treatment (ART) during pregnancy become lost to follow-up (LTFU) during the subsequent year [1–6]. Women who become LTFU are at increased risk of non-adherence to ART and subsequent HIV viremia, increasing the risk of HIV transmission to their infants during pregnancy and breastfeeding [7, 8]. In Eastern Africa, for example, 38% of the 26,000 new HIV infections in children in 2018 were estimated to have resulted from disruptions in HIV care and ART during pregnancy and breastfeeding [1, 9]. A detailed understanding of the factors that influence retention is needed to inform the development of interventions to improve PMTCT outcomes for pregnant and postpartum women living with HIV (PPHIV).

To date, multiple studies have examined retention in the PMTCT continuum in sub-Saharan Africa. These studies have identified a variety of barriers to retention, including younger age, HIV-related stigma and fear of disclosure, low socioeconomic status, inadequate social support, and negative healthcare worker attitude [1, 10–22]. However, many of these studies were conducted prior to the WHO 'Treat All' policy of universal ART eligibility in 2015 [23, 24]. This policy has increased the number of women enrolling in antenatal care who were diagnosed with HIV before, rather than during, their pregnancy [1, 25]. Furthermore, many studies were not conducted in settings in which HIV services were integrated in maternal and child health clinics. These settings may present unique barriers to retention compared to non-integrated settings [26–29]. Such barriers include heightened stigma and fear of public disclosure at clinics providing care to women living with and without HIV, and increased provider workload increasing patient wait times and impacting patient-provider relationships [30].

Finally, most studies assessed factors influencing retention only among PPHIV who were retained in care. Understanding these factors according to retained PPHIV is important because these women may have experienced, and successfully overcome, a variety of barriers to retention, giving them unique perspectives on the relative importance of each factor. However, by not sampling PPHIV who had become LTFU, these studies may not have captured the full range of factors influencing retention in care for this population [31]. The objective of this study was to explore the full range of factors influencing retention in care from the perspectives of PPHIV who were retained in care and LTFU from integrated HIV and maternal and child health services in Kenya.

## Materials and methods

### Setting

Kenya has one of the largest HIV burdens in the world with an HIV prevalence of 6.1% among women 15–49 years of age [32]. Kenya adopted the Option B+ policy in 2014, and in 2016, this policy was subsumed by the WHO "Treat all" policy (i.e., universal test-and-treat strategy for all persons with HIV after diagnosis regardless of age, CD4 count, and pregnancy status) [33, 34]. In addition, routine HIV services were integrated within public maternal and child health clinics in 2015. In this integrated model, HIV services are delivered in antenatal clinics (ANC) and postnatal clinics (PNC) rather than in HIV clinics. These policies have facilitated a significant scale-up of HIV testing and antiretroviral treatment (ART) delivery for general populations and PPHIV in Kenya [35]. As of 2020, an estimated 85% of pregnant women presenting at ANC in Kenya were tested for HIV or knew their HIV status, and 94% of pregnant women living with HIV accessed antiretroviral therapy (ART) for PMTCT [32]. The scale-up of ART

has also helped to drive significant reductions in vertical transmission in Kenya, from 26% in 2009 to 11% in 2018 [36–38].

This study was conducted within the Academic Model Providing Access to Healthcare (AMPATH) in western Kenya. AMPATH is a USAID-funded program that provides HIV care and treatment to more than 170,000 patients at Ministry of Health facilities since 2001 [39]. AMPATH is the largest care program participating in the East Africa International Epidemiology Databases to Evaluate AIDS (EA-IeDEA) Consortium [40]. All clinics provide standard-of-care HIV treatment services based on the Kenyan national guidelines [41].

## Population

Women were eligible to participate in this study if they were enrolled in the EA-IeDEA Kenya PMTCT study, a prospective cohort study that utilized mixed methods to obtain comprehensive data on the factors influencing retention in care for PPHIV and their infants [42]. Eligibility criteria for women were: (i) age ≥18 years, (ii) pregnant or early (i.e. ≤6 months) postpartum, and (iii) enrollment at one of the following AMPATH-affiliated facilities during pregnancy: Busia District Hospital, Huruma Sub-District Hospital, Kitale District Hospital, Moi Teaching and Referral Hospital, and Uasin Gishu District Hospital. These hospitals are a mix of large and small facilities serving urban, peri-urban and rural populations in different ethnic and geographic settings. They are thus representative of the various Ministry of Health facilities in the region. Women not attending the study facility for >90 days and not documented as deceased or transferred out were defined as LTFU.

A purposeful sample of PPHIV enrolled in the prior cohort were selected for qualitative interviews. These women were stratified for selection according to their facility location and care engagement status (retained versus LTFU from the study facility). Women who were retained in care were recruited during their visit to the facility. Women who had become LTFU were recruited in the community after being traced by community health workers. We aimed to sample at least 40 PPHIV, including at least 15 LTFU women, for the interviews. This sample size was estimated to be appropriate for achieving data saturation based on our knowledge of the related literature and our prior qualitative research experience in Kenya, as well as the time and resource constraints of the study.

## Conceptual framework

We developed a conceptual framework based on the Social-Ecological Model and used it across the research process, including to drive data collection and the development of the interview guide and the analysis (Fig 1) [24, 31, 43, 44]. This framework incorporates known barriers and enhancers to retention in the PMTCT literature and is based on the Andersen Behavioral Model of Healthcare Utilization. The Andersen model suggests that peoples' use of health services is related to their predisposition to use services, factors that enable or impede use, and their need for care [31, 43, 44]. In our framework, retention in care must be maintained as women transition from ANC to PNC during the PMTCT continuum, and while confronting various influencing factors in different social-ecological domains. The individual domain encompasses biological and personal history factors such as health beliefs, attitudes and behaviors, perceived and experienced stigma, poverty, and physical and mental health. The peer and family domain encompasses close relationships to the subject including the male partner (hereforth, 'partner') and family members. The community and society domain encompasses societal factors, including social and cultural norms and policies, culture and tradition, and religion. The environment domain encompasses the settings, including the woman's contextual and healthcare environments in which social relationships occur and which

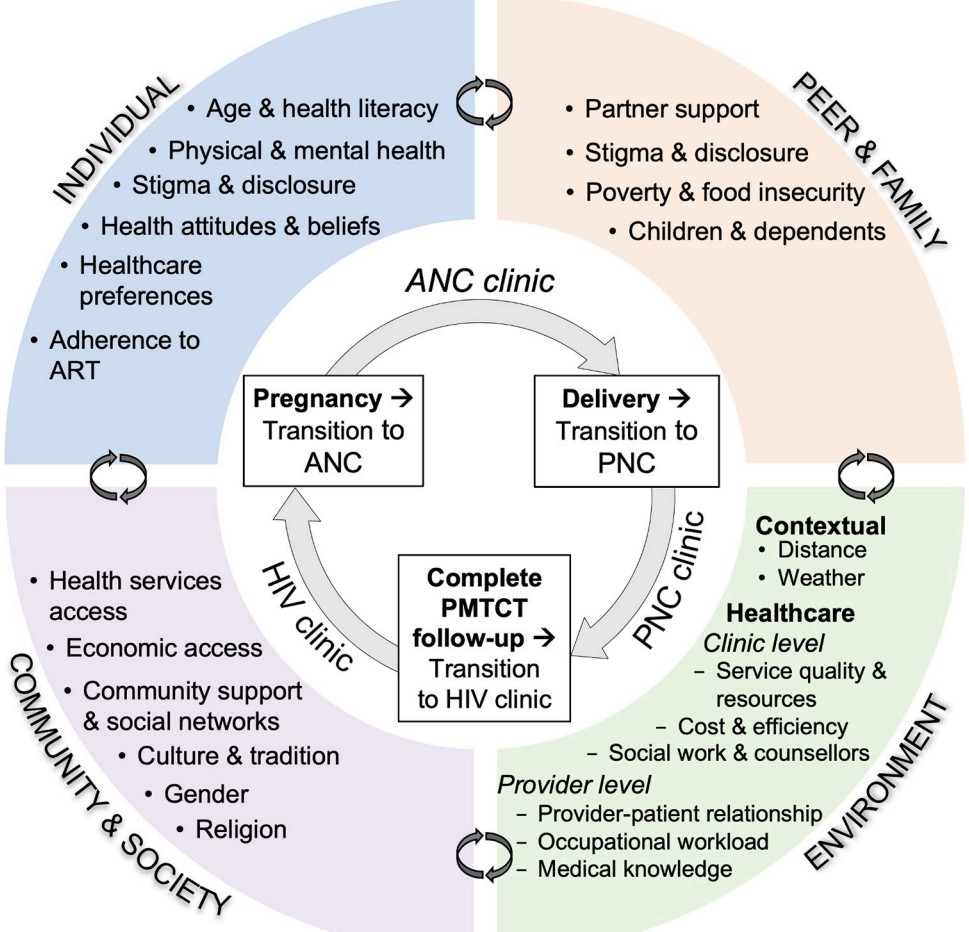

**Fig 1. Social-ecological model of retention in care throughout the PMTCT continuum.**

may structurally influence retention. We hypothesized that the factors influencing retention would be diverse and interdependent, and that differences in perceptions and experiences concerning retention would emerge among retained and LTFU women.

## Data collection and management

Data were collected from participants through in-person interviews with Kenyan research assistants. Demographic and clinical data were collected in REDCap using structured questionnaires. Qualitative interviews were administered using paper-based interview guides and audio-recorded (see S1 and S2 Texts). The interviewer also took field notes on paper relating to context and other elements which could not be captured by audio. The questionnaires and interview guides were designed in English and translated into Kiswahili. Back translation was undertaken to ensure the integrity of the translation. The interviewers were fluent in English and Kiswahili and experienced in conducting qualitative interviews. Separate interview guides were developed for pregnant and postpartum women to account for their differing contexts and experiences.

After adapting the Social-Ecological Model to PMTCT in the Kenyan context, we used it to guide the development of an interview protocol that consisted of semi-structured and open-

ended questions. This ensured that we were exploring the central topics within each domain in the model and identified in the literature, while still allowing for flexibility and room for women to discuss and add their own perceived and experienced barriers and enhancers to retention in care for PPHIV and their infants. Questions were contextualized according to the woman's pregnancy status (i.e., pregnant versus postpartum) and retention status (i.e., retained versus LTFU). Examples of questions include *"What makes it difficult for women living with HIV to attend ANC?", "Why do some postpartum women living with HIV stop attending PNC with their infants?",* and *"Tell me about why you stopped coming to ANC"*. The questionnaire and interview guide were piloted among five PPHIV prior to recruitment.

Interviews were conducted in private settings according to women's preferences and the locations at which they enrolled in the study (i.e., within the clinic for retained PPHIV, and within the participant's home for PPHIV who were LTFU). Each interview lasted 45 to 60 minutes. Participants received US$6 to compensate them for their travel and time. Audio recordings were transcribed and translated verbatim from Kiswahili to English by a trained translator. Identifying details were removed from the transcriptions prior to analysis. Periodic quality checks were also performed by one of the Kenyan investigators, comparing random sections of audio with their corresponding transcriptions and translations to ensure accuracy.

## Analysis

Patient characteristics at enrollment were summarized descriptively using Stata version 10.3. For the analysis, four investigators (J.H., B.K., M.A. and J.W.) began with an open coding framework to identify themes emerging from the data and across interviews. Once a theme was provisionally defined, additional examples were sought during subsequent readings. To ensure validity and reliability, inductive codes were discussed among three investigators (J.H., B.K, and M.A.) to establish a consensus prior to adding them to the codebook. These codes were refined and new codes developed inductively using axial and selective coding of transcripts during the analysis and were informed by our Social-Ecological Model, literature review, and additional themes arising during interviews. Axial coding of each transcript was performed independently by two investigators (J.H. and M.A.), using analytic memos and decision trails to ensure coding consistency and resolving differences through discussion with a third investigator (B.K.). Nvivo 12 was used for the qualitative analysis. Illustrative quotes were selected to highlight key themes that emerged across the data. Data from structured questionnaires and interviews were triangulated to further enhance the construct validity and trustworthiness of the inferences.

## Ethical approval

This study was approved by the Moi University/Moi Teaching and Referral hospital Institutional Research and Ethics Committee in Kenya and the Indiana University Institutional Review Board in the United States. All participants provided written informed consent and the data were coded for analysis using unique study identification numbers assigned to each participant following enrollment, thus ensuring that the data could not be linked back to the participant except through the use of a linkage log.

## Results

### Study participants

From March 2018 to February 2019, 338 PPHIV were enrolled in the study. Among these women, 41 were selected to participate in a qualitative interview (Table 1). The median age

**Table 1. Characteristics of participants at enrollment in the study.**

| Characteristic | Participants |
|---|---|
| | N = 41 |
| | n (%) |
| Age, median years (IQR) | 27 (23–32) |
| Highest level of education | |
| None | 7 (17%) |
| Primary school | 19 (46%) |
| Secondary school | 11 (27%) |
| Tertiary school | 4 (10%) |
| Marital status | |
| Single | 9 (22%) |
| Cohabiting | 30 (73%) |
| Separated or divorced | 1 (2%) |
| Widowed | 1 (2%) |
| Employment | |
| Farmer | 1 (2%) |
| Self-employed | 15 (37%) |
| Unemployed | 19 (46%) |
| Casual worker | 4 (10%) |
| Other | 2 (5%) |
| Facility | |
| Busia District Hospital | 10 (24%) |
| Huruma Sub-District Hospital | 5 (12%) |
| Kitale District Hospital | 10 (24%) |
| Moi Teaching and Referral Hospital | 10 (24%) |
| Uasin Gishu District Hospital | 6 (15%) |
| Travel time to facility, mean minutes (range) | 42 (15–150) |
| Pregnancy status | |
| Pregnant | 29 (71%) |
| Postpartum | 12 (29%) |
| First pregnancy | 7 (17%) |
| Care engagement status | |
| Retained | 25 (61%) |
| LTFU | 16 (39%) |
| Newly positive HIV status | 19 (46%) |
| HIV disclosure status | |
| Disclosed to partner | 30 (73%) |
| No disclosure to anyone | 7 (17%) |

was 27 years, 71% were pregnant and 29% were postpartum, 46% had been newly diagnosed with HIV at enrollment in ANC, and 39% had become LTFU. Among LTFU women, 14 (88%) were disengaged from care (i.e., >90 days since the last visit to any facility) and 2 (12%) had silently transferred to another facility and were in care.1

Women described multiple factors that influenced retention in care in their contexts. Overall, these factors fit within the four domains of our social-ecological model (Fig 1). Some factors were restricted to one domain while others overlapped multiple domains. In some cases, women described enhancers to retention that were the inverse of barriers to retention

described by other women. We therefore use the term 'influencers' to summarize these factors, which may be interpreted as either barriers or enhancers according to their context.

## Individual domain

**Health beliefs, attitudes, and behaviors.**   Within this theme, stories shared by women about their prior knowledge and experiences with PMTCT services were diverse and at times incongruous. Some retained women described that their experience receiving PMTCT services during prior pregnancies enhanced retention during subsequent pregnancies. As one woman who was retained in care (RW) remarked, "*For this it was not a challenge having gone through PMTCT twice.*" In contrast, a woman who had become LTFU (LW) perceived that having a prior successful pregnancy could dissuade women from accessing PMTCT services during a future pregnancy due a self-perceived lack of need for services. She said, "*A person can assume that since she had a successful first pregnancy it is not a must that she goes to the clinic again.*" Lack of experience receiving PMTCT services was also cited as a barrier to retention for women newly diagnosed with HIV. In describing why, one LW who had been newly diagnosed with HIV stated "*After delivery I didn't know where to get medication.*"

Retained and LTFU women expressed differing health beliefs, behaviors and experiences when asked why PPHIV disengaged from care. Retained women more commonly perceived that LTFU women lacked concern for their own and their infant's health. As one RW responded when asked explicitly why a LW would not maintain retention, she said it was because she was "*not being concerned about her health and her baby's [health].*" Another RW qualified this sentiment somewhat, by responding "*There are those who have picked up the habit of stopping clinic after finishing with the main vaccines because they no longer see the need.*" These types of sentiments were not shared by LTFU women. Rather, these women predominantly cited that barriers to retention had to do with fears and experiences around HIV-related stigma and discrimination following unwanted disclosure to others at the clinic, or loss of dignity due to poverty. These themes are described in the next section.

In addition, when probed further about to the barriers to retention, several retained and LTFU women cited the fear of maternal and infant immunizations. One RW said "*Some [postpartum women] are scared of the vaccine injections.*" Mistrust of biomedicine also emerged within this theme, as another LW cited a rumor that postpartum immunizations were secretly co-delivered with family planning medicines. She said, "*Some women say that the injections are doused with family planning drugs.*"

Finally, both retained and LTFU women perceived that the time spent attending ANC during pregnancy was associated with significant opportunity costs when balanced against work responsibilities. When one LW was probed as to why she had disengaged from care during pregnancy, she said:"*It [i.e., attending ANC] has affected my work in so many places. There is a time that I had to go to the clinic every 2 weeks and I was told to go home for treatment and go back only when I am healed completely because I was asking for an off every 2 weeks.*" Retention also impacted home responsibilities. As another RW perceived, "*[Pregnant women] see it [ANC attendance] as a waste of time to go and make a queue while they have chores at home.*" Indeed, for some women, the only value to attending ANC at all was to obtain an 'appointment card' issued at ANC enrollment, which was perceived to enable access to the facility for delivery. This perception is illustrated by one RW's quote: "*I heard some say that you know when you go to give birth they will ask you where your card is and if you do not have a card and they will not serve you at that time. Some women [enroll in ANC] late so that they can get a card. As long as you have a card on the day of giving birth they will not disturb you because they will know you have been going to the clinic.*"

**Stigma and discrimination.** Fear of HIV-related stigma and discrimination at the clinic were prominent barriers to retention, particularly for LTFU women who were newly diagnosed with HIV at ANC enrollment. As one LW stated: "*Many are afraid that if they come to the clinic their [HIV] status will be known.*" Another LW perceived that fear of stigmatization and unwanted disclosure of one's HIV status at the workplace due to frequent clinic visits were barriers to retention. She said, "*If you are pregnant and positive you will have to ask for a day off every month yet you don't want them to know that you are sick. They will ask you why you go to the clinic all the time and that is why some people don't go to the clinic.*"

Fear of diagnosis of HIV was also an important perceived barrier to ANC attendance and delivery at a healthcare facility (versus at home). When asked why some women do not attend ANC, one RW pointed to the impact that an HIV diagnosis has on a woman's life, saying: "*Some really want to go [to ANC] but they are afraid because when you go you must be tested so they will ask 'what will I do if I test positive?' If I am positive, then that is the end of life because we believe that HIV kills and that shock will have a huge impact on you so many don't go to the clinic.*" Women also perceived that fear of diagnosis of HIV in the infant was a barrier to attending clinic. As one RW said, "*Sometimes she is afraid that the baby is also infected so she will be afraid to bring the baby.*" Fear of diagnosis of disease also extended beyond HIV and included other diseases such as syphilis and anemia. As one RW said "*You know when pregnant you get tested for a lot of diseases and that is what people fear that they can be found to be having one so they go to hospitals where they won't be tested.*" Women also expressed fears of abandonment by the partner or other family members following HIV diagnosis or disclosure of their own or their infant's HIV status, as well as anxiety concerning the source of infection and partner's HIV status (see Peer and Family domain).

For LTFU women, fear of stigma and discrimination also included sub-themes that intertwined with ideas about women who possessed other "deeply discrediting" attributes beyond HIV infection, such as those associated with unintended pregnancy, poverty, and the health of the infant [45]. Regarding pregnancy, one LW explained: "*They [i.e., women living with HIV] don't want other people to know that they are pregnant but it will eventually be known because there are people who like minding other people's business.*" Another LW added to this by noting that some mothers feel ashamed in the clinic due to their poverty, which is perceptible to others in the clinic based on their infants' appearance. She said, "*At the clinic you may see a woman hiding her baby. . .and when the baby's weight is taken she become ashamed. At the clinic there are those whose babies are smartly dressed and maybe you earn a little and you have brought yours. . .the baby doesn't even have pampers so it becomes a shame.*" This perception was also echoed by another RW. When she was asked why postpartum women living with HIV might avoid going to clinic, she added that these visits make some mothers feel ashamed because they cannot provide material possessions or nutrition to their infants: "*Maybe it was an unexpected pregnancy so you don't have things like clothes or pampers for the child. Or the baby is sick and his weight has reduced so you will be afraid to go to the clinic because you will have to strip him naked so that he can be weighed and children there are healthy. . .he [the baby] is so small so you don't want people to see him.*"

**Poverty.** Poverty was a pervasive barrier to retention in care for LTFU women. For some women, poverty drove a fear of stigmatization, described above. For other women, poverty posed a direct barrier to accessing the clinic. One LW wept as she explained why she stopped going to clinic after delivery. She said, "*I came in March and then in July, on those other months I didn't come because I didn't have money. Life was just hard. . .money was really not there, if I get some I use it for food.*"

**Physical and mental health of the mother and infant.** A desire to safeguard the infant's health was consistently described by both retained and LTFU women as a key motivator of

retention in care. Preventing HIV transmission to the infant was a major sub-theme. As one RW said in elaborating on why some women engage in ANC early during pregnancy, *"When you come early you will also prevent the baby from those dangers [i.e. HIV disease]."* Women also specifically cited that retention enabled access to ART, which was life-saving. Another RW said *"The one who is positive must take her child [to clinic] because it is that medication [ART] that gives [the child] life."*

Women also described how one's physical health could act as a barrier and enhancer to retention in different circumstances. One RW described how pregnancy-related symptoms could make travel to the clinic difficult, saying *"The pregnancy makes her too tired to even walk, she gets dizzy."* Conversely, poor physical health could also motivate women to attend clinic to access care. Another RW said, *"Being sick will make you come to the clinic."* A woman's mental health also factored into her decision to remain in care. For example, two women who became LTFU after experiencing a miscarriage both described how sadness and anxiety over the loss of their pregnancy contributed to their decisions to disengage from care. As one of these women said, *"If you go there [i.e. to PNC] and see women with children and you don't have yours you will feel bad."* Another woman perceived how emotional distress could be a barrier for women facing an unwanted pregnancy. When asked what may cause pregnant women to disengage from ANC, she replied *"There are times that someone has not accepted that she is pregnant, maybe she is thinking of terminating the pregnancy."* When asked this same question, another LW commented *"There are some women who drink alcohol so they forget about the clinic and remember days later…I have seen many of them."* However, this theme, which did not pertain specifically to PPHIV, did not emerge in the other interviews.

### Peer and family domain

**Partner.** Within the family domain, the influence of the woman's partner was the leading theme. This influence was driven in part by women's economic dependency on their partners, who determined when and where women could access PMTCT services. In our study, 93% of women described themselves as either unemployed, self-employed, or a casual worker (Table 1). When asked why some postpartum women disengage from care, one RW said *"Maybe the father of her child didn't give her money so she won't come [to clinic]."* This influence was partly dependent on the cost of transport, which in turn was dependent on the distance to the facility. Women residing within walking distance of the clinic expressed more autonomy in this regard. The partner's preferences for PMTCT services were also dependent on what seemed to be well-intentioned perceptions about a facility's service quality. When asked why one LW had silently transferred to another facility, she replied *"My husband didn't want me to come here [i.e., the study facility where she initially enrolled in ANC] because of the queue so I attended ANC at Tanaka [another health facility] and delivered there."* Another RW suggested that, similar to some PPHIV, their partners did not want their children receiving immunizations, saying *"Sometimes it is the husbands who refused, they don't want their small babies to go for injections."* The reasons underlying this sentiment were not further explored by the interviewer. Finally, one RW recounted that open communication with the partner and couples HIV testing could have a positive impact on the partner's receptiveness and influence on the woman's health seeking behaviors. She stated: *"They [i.e. partners] are harsh but if you talk to them well there will be an understanding, you can even get tested together. If you start clinic early when you tell him about your status, he will see it wise to also get tested."*

Motivational support from partners was also identified as an influence on retention for PPHIV. This was exemplified by partners providing verbal encouragement and appointment reminders. One LW stated *"If your husband reminds you to go to the clinic on the date written*

*to know how the baby is fairing on, he gives you motivation and morale to go to the clinic.*" The partner could also discourage clinic attendance. When asked why some pregnant women with HIV do not go to clinic, one LW said "*Maybe the husband tells her 'don't go you will go on the day you give birth'.*" None of the women cited attending clinic with their partner as influencing their own or other women's retention in care (<10% of PPHIV attending the study sites were accompanied by their partners at the time of the study). Finally, partner and family violence were cited by several LTFU women as influencers of retention. One LW who was newly diagnosed with HIV during pregnancy recounted her experience after disclosing to her cousin that her infant was HIV-exposed: "*One day I heard the child crying and she [i.e., another family member] told me that my cousin had beaten her so when I asked him why he did that, I received kicks and blows. . .So I thought my uncle would solve that issue but he came and insulted me saying that I should go back to my parents.*"

**Family.** Financial and motivational support from family members also enhanced retention. One RW cited receiving money from her mother to cover the cost of transport to the clinic, saying "*I got to a point where I couldn't work so I couldn't afford money for transport so she used to give me fare and some money to buy even fruits.*" This same woman went on to describe how her father, who was also living with HIV, conveyed support by asking about her child's HIV test result. As she said, "*When the child was tested he was the first one to ask for the results so I felt that he cares more.*"

## Community and society domain

**Culture and tradition.** For some women, attending clinic fulfilled a desire to conform to social norms of motherhood. One LW who had self-transferred to another facility said "*I was happy to go to the clinic because it reaches a time where you just want to give birth so that you can be a mother like others.*" Other women described how the community could be a positive influence on retention by reinforcing the perception that women living with HIV can deliver healthy babies. When asked what motivated one RW to attend clinic, she said "*I heard people saying that there is a [HIV] positive woman who gave birth and the baby is very healthy, I saw the baby and that gave me hope.*" However, community and cultural influences could also be barriers. Some women reported pressure from other women to utilize traditional birth attendants in lieu of facility services. This pressure came particularly from older women who did not utilize similar services in the past. One RW said "*The elderly women will say that in their days they never used to go to the hospital. . .she will tell you that there is no need to go to the hospital.*" Another RW said "*In some cultures, they are not allowed to go to hospitals; they trust their old midwives.*" Pressure to avoid facility-based services could also come from the broader kin group. One LW said "*It depends on the clan that you come from, because you might be married to a clan where they don't allow taking the child to the clinic.*" Rural residence was also linked to lower health literacy, which was in turn perceived as a barrier to accessing facility-based services. One LW who had silently transferred to another facility, said "*Those who live in the remote areas who still don't know the value of hospitals prefer going to midwives.*"

**Mobility.** Mobility, in part related to the cultural tradition in which women traveled to their mothers' home during pregnancy or the postpartum period to receive care from parents or other family members, was another potential barrier to retention. This has been described in Kenya [46] and South Africa [3], and may be influenced by the nature of the study sites as referral facilities with sizeable catchment areas, where some women may prefer to seek longitudinal care at lower-level services closer to their family. One LW who was found to have silently transferred to another facility and was in care, recounted her experience after leaving the study facility: "*I went to different places, I started at Sigoti then I came here [i.e., the study facility] then*

*I had to attend a funeral at home and I attended Sigoti then I visited my husband in Kericho and I went to a clinic there. The date that I was to go for my last clinic is the day I delivered."*

**Religion.** Several women described religion, including ecclesiastical religion and traditional witchraft, as potential barriers to retention. Religious barriers included belief in prayer for healing rather than biomedical healthcare, religious leaders discouraging clinic attendance, and denominations/sects (e.g. Jehova's witnesses) prohibiting the use of medical services and/ or medications. One LW noted that *"There are those who don't take their children to the hospital when they are sick they believe in prayers and that God will heal the children."* Adverse effects from facility-based services (e.g., an infant's leg swelling at a vaccine injection site) were also linked to witchcraft. As one LW said *"There are those who believe in witchcraft so they feel if they go for the injections then they will be affected."*

### Environment domain

**Contextual environment.** In the contextual environment, residing in a rural area or further from the clinic was viewed as a barrier to retention given the increased cost and difficulty of transport. As stated by one LW "*It depends with the distance, not all of them live near the hospital and maybe they don't have money for transport so they will skip this date and go on the other one and they keep postponing.*" She went on to say "*You might not have money for a motorcycle and you can't walk with the baby so you will wait until you get money.*" Another woman expressed difficulty accessing the facility during labor because private-hire motorcycle drivers were unwilling to take them for fear they will deliver on the way. Adverse weather conditions, including cold weather and rain, were additional barriers to attending the clinic, particularly for postpartum women carrying infants. One such woman said "*Sometimes it may be raining the whole day so you just cancel the appointment.*"

**Healthcare environment—Clinic sub-level.** Within the clinic sub-level, poor service quality was a prominent barrier to retention. When one LW who had delivered at home was asked why women like her did not deliver at the hospital, she replied "*We are a little bit not happy with the hospital because the maternity beds are small yet three women are made to share one so the children will sleep on the bed while the mothers sleep on the floor or while sitting.*" Women also described accessing different care venues based on the types and costs of services they offered. As one RW said, "*There are those who believe in Gogo [i.e., traditional midwife] so after delivery they will take their children to Gogo who treats them with traditional herbs. They will take the children to clinic for vaccines but for any other ailment they take them to Gogo.*" The queue was a recurring theme that reflected facility service quality for both retained and LTFU women, who cited long queues and waiting times spent at the clinics as barriers to retention. As one LW said, "*If you go to the clinic you are just making yourself tired for nothing and you are going to find a long queue; just decide to go to the midwife it will just take you five minutes.*" Private clinics, compared to public clinics, were predominantly perceived to offer more efficient services to patients who could afford them. As one LW said "*Some go to Alupe or another private [clinic] because they know that they will pay, you will take 10–30 minutes but at the government hospitals there are no bribes you just follow the queue so you can stay there the whole day.*"

The configuration of PMTCT service delivery also influenced retention. On one hand, integration of routine HIV services into maternal and child health clinics was perceived to ease care access for women during pregnancy compared to non-integrated services. As one LW said, "*When I was pregnant I came to the pregnancy clinic and the [HIV clinic] used to be at different times. So when they came at different times it was a little bit hard for me.*" On the other hand, integrated services were perceived to increase the risk of accidental HIV status disclosure

to other patients attending the clinic. This was because integrated clinics were perceived to lack space and privacy. This configuration also dissuaded PPHIV from attending the clinic with other women in their community as they would normally do. As one RW said "*Advantage of the HIV clinic for those who are not in denial you don't feel stigmatized because those attending to you know the type of people that they are attending to compared to this place [i.e., the postnatal clinic providing integrated services].*" The increased frequency of clinic appointments given to PPHIV, compared to women who were not living with HIV, were also viewed to increase the risk of disclosure, and discourage early enrollment in ANC. When asked about this, one RW replied *"When you start late the visits will be few, you go for only two months and deliver. If you start early you will have to attend every month and that is why people start as seven months. . .women are scared of many visits, that is what I also feared."* Some women reported that early engagement in care during pregnancy was also discouraged at the community level for this reason, and was tied to the individual-level perspective about the limited utility of ANC services that some women had.

The cost of care was also an important influencer on women's healthcare seeking practices and was intertwined with the individual-level theme of poverty. One RW explained how costs could be a barrier to attending ANC by stating that *"Even if the public [facility] is free there are some things that you pay for and she can't get that money so she just stays at home."* One RW elaborated on this concept by noting that poverty could motivate delayed engagement in care during pregnancy, saying *"Some think that if they start [clinic] early they will use a lot of money to even get tested and they don't have it."* Another RW commented that the cost of facility-based services could motivate women to seek alternative care venues, such as a midwife. She said, *"When you start clinic you must have extra money and maybe you have nothing so she stays at home and deliver at the midwife where you will only pay 100 or 200 shillings."* It was apparent that some women's healthcare seeking behaviors were influenced by the cost of services, and that this influence could be dynamic during pregnancy and the postpartum period. As one LW remarked, *"Some prefer going to private clinics and some go to public. Maybe you chose a private one while pregnant and after delivery you have to go with money yet you don't have."*

**Healthcare environment—Provider sub-level.** Within the provider sub-level, retained and LTFU women expressed that providers who were welcoming and demonstrated empathy and encouragement enhanced retention. For some women, this helped them overcome other structural barriers to retention, such as long queues. As one RW said "*At AMPATH I have a very good friend. . .who works at the lab. When I go there, [he] always buys me tea, he knows I don't like making a queue. . .and that made it easy.*" Conversely, negative provider attitudes and interactions were viewed as barriers to retention. One LW recounted her experience attending clinic, saying "*You make a queue and then you find the provider is in a bad mood and she can even insult you and you start regretting going there.*" When asked why one LW did not return to clinic after missing her scheduled appointment, she said "*I was scared because I knew if I come they would quarrel me so I didn't.*" Finally, several women reported that negative feedback from providers discouraged others from enrolling early in care during pregnancy. One LW recounted her experience with this, saying "*I was told [by a provider] 'you have a small stomach you are just two months along and you are already going to the clinic you will get tired by the time you get to 8 or 9 months.*' This quote also suggests that the provider recognized the challenge that frequent clinic appointments posed for PPHIV.

## Discussion

This study demonstrates that the barriers and enhancers to retention in care for PPHIV are multitudinous, complex and interrelated across different social-ecological domains. This study

also illustrates the importance of sampling patients who have become LTFU when formatively assessing retention, as several themes emerged from this group that may not have been sufficiently captured if only retained women were sampled. LTFU women in general expressed diminished agency compared to retained women. This was primarily driven by perceived and experienced stigma and fears of discrimination resulting from disclosure of their HIV status. This finding is supported by prior research in Kenya and the surrounding region [47, 48]. This finding also underscores the importance of interventions to reduce barriers stemming from concerns about HIV-related stigma and disclosure. This includes interventions to enhance partner involvement in PMTCT, as well as psychosocial interventions to address mental health, stigma and domestic violence [48, 49].

Retained women, in contrast to LTFU women, emphasized how their healthcare seeking behaviors were largely influenced by factors relating to healthcare service quality and efficiency. These themes underscored the sense of agency that retained women had concerning what they perceived to be a market-oriented healthcare system in their setting. Further research utilizing behavioral economics approaches could help care programs optimize service delivery to better match retained patients' preferences as consumers of healthcare services [50].

Yet while some PPHIV view the healthcare system in western Kenya as a market-oriented economy in which programs must compete to offer the most desirable services, this notion also implies that services are only available to those who have the means to access them [51]. Indeed, many of the barriers to retention in our study represent social determinants of health that will be challenging for healthcare programs to influence. However, several structural barriers within the healthcare environment suggest opportunities for interventions. For example, retained and LTFU women perceived that integrated services increased the risk of HIV status disclosure. All of the facilities in our study had a common reception area and separate clinical areas for HIV-positive and negative women, respectively. Thus, it is possible that patients could identify one another's HIV status based on their triage path within the clinic. Reconfiguring these clinics to better protect patient privacy by using private clinic rooms that serve both HIV positive and negative clients could enhance retention for some PPHIV. This may not be feasible at some clinics, however. The ANC and PNC sites used for recruitment in our study were nested within public referral facilities and had limited space for triage, patient encounters, recordkeeping, and other ancillary services which would not be easily reconfigured to enhance patient privacy. This highlights a major challenge of integration in resource-constrained clinics, which may not have sufficient capacity to handle the additional service needs of many PPHIV in high HIV burden settings. To reduce the risk of attrition, some women may be or offering women a choice to receive integrated or non-integrated services, particularly if their partner is also living with HIV, which may also help to promote male engagement in PMTCT and family-centered care.

Improving the efficiency of service delivery could also enhance retention. Multiple women, both retained and LTFU, perceived that long queues and prolonged time spent at the clinic outweighed the utility of the services to them, particularly given competing priorities at their homes and workplaces. Differentiated service delivery models have recently been developed to address this issue among non-PMTCT clients [41, 52]. These models simplify and expedite HIV service delivery for clients who are clinically stable (i.e., those who are able to maintain retention in care and viral suppression), allowing programs to refocus resources on unstable clients (e.g., those who are at risk of becoming LTFU or who are not virally suppressed). Differentiated service delivery models for PPHIV are lacking, however, and research is needed to understand the adaptation of this model to the PMTCT population [41, 53, 54]. These models may also help reduce the risk of stigma and disclosure at integrated clinics by lessening

appointment frequencies and durations for PPHIV, aligning them more closely with HIV negative women and minimizing these elements as distinguishing characteristics between both groups [30].

Our study has strengths and limitations. Incorporating the perspectives of LTFU women is a unique strength that few studies have undertaken, and which has enhanced the richness of our data. Participants were also sampled from five facilities across western Kenya, enhancing the generalizability of the results. The inclusion of pregnant and postpartum women is also a strength, as the experiences and perspectives of women attending ANC and PNC clinics may differ. However, our study did not include women >6 months postpartum, who may have had additional perspectives on retention that were not captured in our study. The same is true for women's partners and healthcare providers.

## Conclusion

Multiple social-ecological factors influence retention in care for PPHIV in Kenya. Stigma, concerns about unwanted HIV status disclosure, and poverty are prominent barriers to retention for LTFU women. Retained women, in contrast, expressed themes relating to healthcare service quality and efficiency that influenced their healthcare seeking behaviors. PPHIV must navigate multiple, sometimes conflicting social roles, statuses and identities related to HIV, motherhood, and social class, to maintain retention in care. Interventions to address multiple barriers simultaneously will have the greatest impact given the complex nature of retention for this population. Such interventions will need to be highly contextualized and individualized. HIV care programs offering integrated HIV and maternal and child health services should also carefully assess the configuration of the clinics in which integrated services are being delivered, and the ways in which it may enhance or diminish retention in care for PPHIV. These issues represent critical challenge for care programs and policy organizations given the magnitude of non-retention in the PMTCT continuum in sub-Saharan Africa.

## Supporting information

**S1 Text. Interview guide for pregnant women, English and Kiswahili versions.**
(DOCX)

**S2 Text. Interview guide for postpartum women, English and Kiswahili versions.**
(DOCX)

## Acknowledgments

We thank the research assistants, clinical staff and community health workers at each of the study sites for their roles in facilitating recruitment and data collection.

## Author Contributions

**Conceptualization:** Marsha Alera, Bett Kipchumba, Julia Songok, Winfred Mwangi, Constantin Yiannoutsos, Juddy Wachira, Kara Wools-Kaloustian.

**Data curation:** John Humphrey, Beverly Musick.

**Formal analysis:** John Humphrey, Marsha Alera, Bett Kipchumba, Beverly Musick, Juddy Wachira.

**Funding acquisition:** John Humphrey, Bett Kipchumba, Constantin Yiannoutsos, Kara Wools-Kaloustian.

**Investigation:** John Humphrey.

**Methodology:** Kara Wools-Kaloustian.

**Software:** Beverly Musick.

**Supervision:** Elizabeth J. Pfeiffer, Julia Songok.

**Writing – original draft:** John Humphrey.

**Writing – review & editing:** Marsha Alera, Bett Kipchumba, Elizabeth J. Pfeiffer, Julia Songok, Winfred Mwangi, Juddy Wachira, Kara Wools-Kaloustian.

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
