## [Decision Letter · Decision Letter 0]

7 Jul 2021

 PGPH-D-21-00098 A qualitative study of the barriers and enhancers to retention in care for pregnant and postpartum women living with HIV: the EA-IeDEA Kenya PMTCT study PLOS Global Public Health

Dear Dr. Humphrey,

Thank you for submitting your manuscript to PLOS Global Public Health. After careful consideration, we feel that it has merit but does not fully meet PLOS Global Public Health’s publication criteria as it currently stands. Therefore, we invite you to submit a revised version of the manuscript that addresses the points raised during the review process.

We look forward to receiving your revised manuscript.

Kind regards,

Mishal S. Khan, PhD

Academic Editor

Journal Requirements:

Additional Editor Comments (if provided):

Reviewers' comments:

Reviewer's Responses to Questions

**Comments to the Author**

1. Does this manuscript meet PLOS Global Public Health’s publication criteria? Is the manuscript technically sound, and do the data support the conclusions? The manuscript must describe methodologically and ethically rigorous research with conclusions that are appropriately drawn based on the data presented.

Reviewer #2: Partly

2. Has the statistical analysis been performed appropriately and rigorously?

Reviewer #2: N/A

3. Have the authors made all data underlying the findings in their manuscript fully available (please refer to the Data Availability Statement at the start of the manuscript PDF file)?

Reviewer #2: Yes

4. Is the manuscript presented in an intelligible fashion and written in standard English?

Reviewer #2: Yes

5. Review Comments to the Author

Reviewer #2: This is an interesting contribution, which provides useful insights and lessons to better understand the barriers to retention in care of pregnant and postpartum women living with HIV.

The paper is well organised and well written. The presentation of findings is clear and well structured around four levels of analysis, from the individual level to the wider health care context. The inclusion of lost to follow-up participants is particularly valuable.

I have a few suggestions the authors could consider when revising the manuscript:

- In the introduction the authors write that “many studies were not conducted in settings in which HIV services were integrated in maternal and child health clinics. These settings may present unique barriers to retention compared to non-integrated settings”. Can you specify and elaborate on these barriers? This would be useful to further highlight the value of the paper.

- The description of the setting is thin and insufficient to clarify the policy context. I suggest the authors provide more details and explanation about the “Treat All” policy and its implementation in Kenya – when, what, who, and how.

- Why were 41 participants interviewed? This is a reasonable sample size, but I wasn’t sure how you reached that number. Did you sample until data saturation? Or was the sampling size determined by time and resource constraints?

- The conceptual framework is sound. I particularly liked the way in which the four domains are used (ie individual, family, community, health system level) to organise the presentation of findings. It is also good you recognised that some themes such as stigma are cross cutting. However, I wonder how the framework in Figure 1 was used in the research process. Was it developed to guide data collection, data analysis, or both? I am also not so sure Figure 1 is really necessary. Perhaps that figure could be refined in light of the findings (and placed in the result section or at the beginning of the discussion).

- In the methods section, the authors mention that "new codes were developed inductively using axial and selective coding". However, there is no reference to open coding (which is the foundation of inductive analysis).

Minor points:

- The section on “mobility” should be revised. In particular, the following statement needs explanation: “the cultural tradition in which women traveled to their mothers' home during pregnancy or the postpartum period, was another potential barrier to retention”.

- The conclusion in the abstract could be more forceful

- HIV is ok in the title but less known acronyms such as EA-IeDEA and PMTCT should not be used

Thank you for giving me the opportunity to comment on this interesting paper.

6. PLOS authors have the option to publish the peer review history of their article (what does this mean?). If published, this will include your full peer review and any attached files.

**Do you want your identity to be public for this peer review?** For information about this choice, including consent withdrawal, please see our Privacy Policy.

Reviewer #2: No

**Comments to the Author**

1. Does this manuscript meet PLOS Climate’s publication criteria? Is the manuscript technically sound, and do the data support the conclusions? The manuscript must describe methodologically and ethically rigorous research with conclusions that are appropriately drawn based on the data presented.

Reviewer #1: Yes

2. Has the statistical analysis been performed appropriately and rigorously?

Reviewer #1: N/A

3. Have the authors made all data underlying the findings in their manuscript fully available (please refer to the Data Availability Statement at the start of the manuscript PDF file)?

Reviewer #1: No

4. Is the manuscript presented in an intelligible fashion and written in standard English?

PLOS Climate does not copyedit accepted manuscripts, so the language in submitted articles must be clear, correct, and unambiguous. Any typographical or grammatical errors should be corrected at revision, so please note any specific errors here.

Reviewer #1: Yes

5. Review Comments to the Author

Reviewer #1: The paper is focused on very relevant topic - retention in care for both pregnant and postpartum women living with HIV (PPHIV), through the lenses of the maternal and childcare services in Kenya. The effort to include both PPHIV who are retained in services, as well as ones who are lost to follow-up (LTFU) is commendable.

Considering the fact that LTFU women have an opportunity to voice the reasons that hampered for them, and their children, access to health and HIV services, authors should justify what is the rationale for offering in parallel ‘second -hand’ perceptions of women retained in care about LTFU ones.

The Discussion (and consequently conclusion) would benefit from further refection on how integrated PMTCT services, at the sites were participants in the study were recruited, are organized and where potential changes/ improvements can be introduced to minimize lost to follow up .

6. PLOS authors have the option to publish the peer review history of their article (what does this mean?). If published, this will include your full peer review and any attached files.

**Do you want your identity to be public for this peer review?** For information about this choice, including consent withdrawal, please see our Privacy Policy.

Reviewer #1: No

---

## [Editor Report · Decision Letter 1]

15 Sep 2021

A qualitative study of the barriers and enhancers to retention in care for pregnant and postpartum women living with HIV

PGPH-D-21-00098R1

Dear Dr. Humphrey,

We're pleased to inform you that your manuscript has been judged scientifically suitable for publication and will be formally accepted for publication once it meets all outstanding technical requirements.

Within one week, you'll receive an e-mail detailing the required amendments. When these have been addressed, you'll receive a formal acceptance letter and your manuscript will be scheduled for publication.

An invoice for payment will follow shortly after the formal acceptance. To ensure an efficient process, please log into Editorial Manager at https://www.editorialmanager.com/pgph/ click the 'Update My Information' link at the top of the page, and double check that your user information is up-to-date. If you have any billing related questions, please contact our Author Billing department directly at authorbilling@plos.org.

Kind regards,

Mishal S. Khan, PhD

Academic Editor
